# The Impact of Consuming Zinc-Biofortified Wheat Flour on Haematological Indices of Zinc and Iron Status in Adolescent Girls in Rural Pakistan: A Cluster-Randomised, Double-Blind, Controlled Effectiveness Trial

**DOI:** 10.3390/nu14081657

**Published:** 2022-04-15

**Authors:** Swarnim Gupta, Mukhtiar Zaman, Sadia Fatima, Babar Shahzad, Anna K. M. Brazier, Victoria H. Moran, Martin R. Broadley, Munir H. Zia, Elizabeth H. Bailey, Lolita Wilson, Iqbal M. Khan, Jonathan K. Sinclair, Nicola M. Lowe

**Affiliations:** 1Lancashire Research Centre for Global Development, University of Central Lancashire, Preston PR1 2HE, UK; sgupta6@uclan.ac.uk (S.G.); akmbrazier1@uclan.ac.uk (A.K.M.B.); vlmoran@uclan.ac.uk (V.H.M.); jksinclair@uclan.ac.uk (J.K.S.); 2Department of Pulmonology, Rehman Medical Institute, Peshawar 25000, Pakistan; mukhtiar.zaman@rmi.edu.pk; 3Institute of Basic Medical Sciences, Khyber Medical University, Peshawar 25100, Pakistan; drsadiafatima@gmail.com (S.F.); babar.kmu@gmail.com (B.S.); 4School of Biosciences, Sutton Bonington Campus, University of Nottingham, Leicester NG7 2RD, UK; martin.broadley@nottingham.ac.uk (M.R.B.); liz.bailey@nottingham.ac.uk (E.H.B.); lolita.wilson@nottingham.ac.uk (L.W.); 5Rothamsted Research, West Common, Harpenden AL5 2JQ, UK; 6Research and Development Department, Fauji Fertilizer Co., Ltd., Rawalpindi 46000, Pakistan; munirzia@ffc.com.pk; 7Department of Pathology, Rehman Medical Institute, Peshawar 25000, Pakistan; iqbal.muhammad@rmi.edu.pk

**Keywords:** zinc, biofortification, plasma zinc concentration, adolescent girls, iron status, wheat flour, Pakistan, minerals, deficiency

## Abstract

Biofortification of wheat is potentially a sustainable strategy to improve zinc intake; however, evidence of its effectiveness is needed. A household-based, double-blind, cluster-randomized controlled trial (RCT) was conducted in rural Pakistan. The primary objective was to examine the effects of consuming zinc-biofortified wheat flour on the zinc status of adolescent girls aged 10–16 years (*n* = 517). Households received either zinc-biofortified flour or control flour for 25 weeks; blood samples and 24-h dietary recalls were collected for mineral status and zinc intake assessment. Plasma concentrations of zinc (PZC), selenium and copper were measured via inductively coupled plasma mass spectrometry and serum ferritin (SF), transferrin receptor, alpha 1-acid glycoprotein and C-reactive protein by immunoassay. Consumption of the zinc-biofortified flour resulted in a moderate increase in intakes of zinc (1.5 mg/day) and iron (1.2 mg/day). This had no significant effect on PZC (control 641.6 ± 95.3 µg/L vs. intervention 643.8 ± 106.2 µg/L; *p* = 0.455), however there was an overall reduction in the rate of storage iron deficiency (SF < 15 µg/L; control 11.8% vs. 1.0% intervention). Consumption of zinc-biofortified flour increased zinc intake (21%) but was not associated with an increase in PZC. Establishing a sensitive biomarker of zinc status is an ongoing priority.

## 1. Introduction

The United Nations Sustainable Development Goals for zero hunger (SDG2) and good health and wellbeing for all (SDG3) cannot be accomplished without alleviating micronutrient deficiencies, often referred to as ‘hidden hunger’, which continues to be a global public health challenge [1]. Micronutrient deficiencies, especially zinc and iron, affect more than a billion people worldwide with the low- and middle-income countries (LMICs) carrying the greatest burden by far [2,3,4]. Functional consequences of zinc deficiency are well known and include compromised physical growth, immune competence, neurobehavioral and cognitive development, as well as complications during pregnancy and childbirth [5,6]. In LMICs, a concomitant high prevalence of poor diet (which is low in micronutrient content and/or poor bioavailability) and infectious disease form a negative cycle impacting educational attainment, economic efficiency, and national development.

In Pakistan, 22.1% of women of reproductive age (WRA) and 18.6% of children under five years of age are zinc-deficient [7]. This prevalence of zinc deficiency, based on low plasma/serum zinc concentration (PZC), varies not only by region but also by rural or urban residence (24.3% vs. 18.7% for women), with a greater prevalence in rural areas. The situation is exacerbated in more marginalized communities where poverty, low education levels, poor infrastructure and access to health care are contributing factors. Previously, we reported the prevalence of zinc deficiency among women of reproductive age (WRA) in a marginalised community near the brick kilns close to Peshawar in Khyber Pakhtunkhwa province as 30%, based on a plasma zinc concentration (PZC) cut-off 660 µg/L [8]. Stunting (another recommended indicator of population zinc status) among children under five years of age is as high as 40.0% in Khyber Pakhtunkhwa province and 48.3% in the KP-NMD (Khyber Pakhtunkhwa’s newly merged districts), which is the highest prevalence rate in the country [7]. Based on secondary data analysis from previous national-level surveys, the Global Alliance for Improved Nutrition (GAIN) reports high levels of stunting (22%) and plasma zinc deficiency (42%) among adolescent girls aged 15–19 years in Pakistan [9]. Since adolescence is a critical developmental phase of the human lifecycle, such deficiencies not only negatively impact the health of these girls, but also the health of their future offspring and, ultimately, the community.

Strategies to improve zinc nutrition include supplementation, diet diversification, fortification and biofortification. In LMICs such as Pakistan, zinc supplementation is often targeted where a need is identified or limited to the therapeutic purpose of treating diarrhoea rather than as a preventive strategy mainly because of financial and logistical reasons. Diet diversification is still in the nascent stages of its development in the absence of programmatic experience with the promotion of home-processing techniques to increase absorbable zinc in the diet [10]. It also requires changes in behaviours and food choices that are often deeply embedded in culture and identity or unachievable due to food insecurity. Food fortification is gaining attention in LMICs. In Pakistan, a food fortification program to fortify wheat flour with iron and folate, and edible oil/ghee with vitamin A and D has already been rolled out. Based on the mid-term evaluation of this program, flour fortification was less successful compared to oil due to the significant challenges in ensuring the consistency of fortification levels, tensions between the programme and the industry association, the discolouring effect of fortified flour on baked food items, the absence of government inspection and enforcement of mandatory legislation, and lack of consumer demand [11]. Targeted biofortification using traditional plant breeding and/or agronomic practices appears to be promising for reducing zinc deficiency in LMICs because it is potentially sustainable and highly cost-effective. It can cater to the communities that are geographically hard to reach; who subsist primarily on staples because of the unaffordability of a diverse diet; where food is grown, processed, and consumed locally, therefore bypassing centralized processing mills for fortification—a phenomenon common in Pakistan. Conventionally bred zinc-rich varieties, such as Zincol-2016, and more recently, Akbar-2019 developed by Harvest Plus and partners, have already been released in Pakistan. However, there is a paucity of evidence from large-scale studies to investigate the effectiveness of the zinc biofortified crops in improving health outcomes in free-living community settings.

Biofortified Zinc Flour to Eliminate Deficiency (BiZiFED) is a research programme that aims to generate data on the effectiveness, acceptability, and feasibility of a wheat biofortification strategy for the improvement of micronutrient status in Pakistan. Our previous foundation study under this programme, which comprised a smaller efficacy trial with a randomised cross-over placebo-control design in women of reproductive age (*n* = 50), showed regular biofortified flour consumption for eight weeks increased the daily dietary zinc intake by 30–60% depending on the bran content of the flour [12].

BiZiFED2 is a broad transdisciplinary study in which one of the components is a randomised controlled trial (RCT) that investigates the effectiveness of consuming flour milled from biofortified high zinc wheat variety (Zincol-2016 grown with zinc fertilisers) on biochemical and functional indices of zinc and iron status in adolescent girls and children living in a low-resource free-living setting in Khyber Pakhtunkhwa. There are many regions of Pakistan where, for physical geolographical reasons, the soil has low levels of plant-available zinc [13], leading to low levels of zinc accumulation in the grain of wheat, one of the staple crops. Therefore, the BiZiFED2 programme employed agronomic techniques to optimise the zinc uptake by a conventionally bred high zinc variety wheat crop, Zincol-2016 [12].

In this paper, we present the impact of consuming Zincol-2016 grown with zinc fertilisers (henceforth referred to as ‘biofortified’ wheat in the context of this study) on zinc intake and haematological indices of micronutrient status of adolescent girls. The primary objective was to examine the impact of consuming zinc biofortified flour on plasma zinc concentration as an indicator of zinc status in adolescent girls. Our earlier study revealed that Zincol-2016-biofortified grain had both a greater zinc and iron content compared to control grain; therefore, the iron status was included as a secondary outcome measure for the present study. Other secondary objectives included the quantification of the contribution to total dietary zinc and iron from the biofortified flour, and the impact of consuming biofortified flour on plasma copper and selenium concentration and inflammatory biomarkers.

## 2. Materials and Methods

### 2.1. Setting, Participant Recruitment and Study Design

We conducted a double-blind, placebo-controlled, cluster randomised, household-based trial in two neighbouring catchment areas of rural Khyber Pakhtunkhwa, Pakistan, between November 2019 and March 2021. Ethical approval was granted from the University of Central Lancashire STEMH Ethics Committee (reference number: STEMH 1014) and Khyber Medical University Ethics Committee (reference number: DIR/KMU-EB/BZ/000683). The study was registered with the ISRCTN registry (Trial registration number ISRCTN17107812). A cluster randomised trial was necessary due to the community-based nature of the intervention and to reduce the possibility of contamination as sharing flour with neighbouring households or consuming meals together is a regular practice. Reporting of this study complies with the Consolidated Standards of Reporting Trials (CONSORT) guidelines for cluster randomised analyses [14].

The complete protocol, including the study setting, cluster selection and randomisation, subject recruitment and consent process, and estimation of sample size, has been published in detail [15]. However, some deviation from the planned duration of study phases because of the COVID-19 pandemic is presented here (Figure 1). Briefly, two geographically proximal catchment areas A (5 km^2^) and B (4.5 km^2^), located 30–40 km southeast of Peshawar, comprising 23 and 21 clusters (hamlets), respectively, were assessed for eligibility to participate in the study in June 2019. Eligibility criteria were defined as households with at least one unmarried, non-pregnant, non-lactating adolescent girl (10–16 years) and one child (1–5 years). There were no additional inclusion or exclusion criteria. Household eligibility was recorded along with the household size (number of individuals cooking and eating together) and both were used in the cluster selection and randomisation process.

Clusters were arranged in order according to the mean eligible household size, starting with those with the smallest average family size. They were then sequentially included until the target sample size of 500 adolescent–child pairs was reached. This sample size was estimated to detect a 48 µg/L change in plasma zinc concentration based on the previous study and: (1) a pragmatic and conservative estimate of the likely intraclass correlation coefficient (ICC) accounting for a moderate ICC value (ICC = 0.1) as recommended [16]; (2) a pragmatic estimate of the expected average cluster size based on discussions with the Pakistan field team leader (*n* = 15); (3) to account for unequal cluster sizes an estimate of the coefficient of variation of cluster sizes (0.55) was calculated using the equation of Eldridge et al. [17] taking into account the expected mean cluster size as well as the expected range of cluster sizes, similarly based on discussions with the Pakistan field team; (4) a significance level at 5% (two-sided) and 90% power with an attrition rate of 20%.

This cluster selection process resulted in 28 clusters composed of 483 eligible households with an equal distribution of 14 clusters in each catchment area, which was coincidental as the selection was not dependent on the cluster location but solely on family size. To assess willingness to participate in the study, a follow-up survey of all households within the included clusters was undertaken in September 2019 using a complete enumeration method where the head of each household was approached, and the purpose of the study explained. If the head of the household agreed, the adolescent girl and the mother of the child were approached and the purpose of the study explained, along with the participant information, which was provided in written form in the local language (Pushto) and explained verbally. Consent to participate was indicated by signing with initials or an ‘X’ on the consent form. In households where there were two or more eligible adolescents and children, all pairs were invited to participate. Some households withdrew (*n* = 160) their consent at the time of enrolment, therefore, three additional clusters from each area were selected based on the mean household size [15]. Finally, 517 adolescent/child pairs from 486 households located in 34 clusters were recruited and enrolment data collected. These 34 clusters were matched into pairs according to the average household size of the cluster and the age of the participating adolescent girl to help create a comparable baseline for control and intervention arms [18,19]. Within each pair of clusters, allocation to the intervention or control arm of the study was done randomly via a computer-generated software application (Random Allocation Software) so each cluster had an equal prior probability to receive the intervention flour or control flour during the intervention phase (phase II of the study). A planned baseline phase of 6 months (phase I) started on 4 November 2019 where participating households in all study clusters were provided with wheat flour milled from locally purchased grain. Towards the end of this phase, baseline data were collected for the outcome measures. This initial six-month period was intended to allow stabilization of any fluctuations caused by introduction of this additional ‘food income’, such as any changes in their usual dietary patterns, and to help establish a robust baseline prior to the initiation of the intervention phase (phase II). However, to safeguard the communities and project staff during the first COVID-19 wave, which coincided with phase I, this stabilisation period was extended to 10.5 months. The intervention phase (phase II) began on 22 September 2020, with baseline data collected immediately prior to this in late August and early September. In phase II, all participating households were provided with either the control (Galaxy variety) or the biofortified flour (Zincol-16) depending on the arm of the study they belonged to and was continued for 25 weeks (until mid-March 2021).

Participants were contacted by the field staff at five time points, T1 to T5, for collection of samples and data. Two of the time points (T1 and T2 at the beginning and middle of phase I) were a part of the stabilization period and hence data for these has not been considered for the purpose of assessing the impact of the intervention. To measure the effect of the intervention, baseline data for primary and secondary outcomes were collected towards the end of phase I (T3; prior to the initiation of phase II) and was followed by another two subsequent rounds of data collection: at mid (T4) and endpoint (T5) of the second phase of the study.

During the entire intervention phase, sufficient freshly milled flour for all household members (calculated based on the household size and the flour consumption rate reported during recruitment) was supplied every 15 days. Participants collected the flour from a distribution point located in each area (A and B), on presentation of a voucher.

Compliance was monitored throughout the study via through multiple spot checks (Median 5, range 2–9) of the majority of participating households (*n* = 399) during household visits by the project staff. Staff noted if the study flour was present and being consumed and asked if additional flour had been purchased. In addition, in order to receive each 15-day flour supply, the family member visiting the distribution point had to exchange the bag from the previous supply.

All batches of distributed flour were sampled to monitor the iron and zinc content. During the entire study, households were encouraged to restrict the use of the study flour to members of their own household.

### 2.2. Masking

The study was double-blind throughout the trial and data were analysed using alphabetical codes for the identification of the two arms. Only one member (MZ), who was not directly involved in the data collection or analysis but supervised logistics of flour distribution, performed the cluster randomisation to the intervention or control arm and had access to the recorded allocation for each cluster. The participants and all the research team members, including the principal investigator, co-investigators (except MZ), field team, laboratory scientists, database manager and data analysts, remained blinded to the allocation of the clusters until the primary outcome analysis was complete.

The two varieties of grains that were used in the intervention phase were stored within the same building in two separate storage rooms coded ‘Store A’ and ‘Store B’. To keep the personnel involved in processing and distribution blinded, it was ensured that the two varieties of grain arrived in the store at the start of the study at the same time. To avoid contamination, batches of the two varieties of grains were milled on separate days in a local mill contracted for this purpose and returned to the respective stores. All the processes in the mill were supervised by the study logistics officer. The sacks containing both varieties were identical and labelled using a unique alphanumeric code, which did not disclose the type of flour it contained but could be used to monitor adherence and fidelity of treatment allocation. Return of empty sacks from the previous supply by the household member were reconciled with the codes on the distribution list to ensure the right bag was provided to the household.

### 2.3. Local Production of Zincol-2016 Grain for the Intervention

Zincol-2016 is a relatively new variety of wheat that was released in Pakistan in 2016. This variety was bred by HarvestPlus and its local partner, National Agriculture Research System Pakistan, to achieve a target grain zinc concentration of 37 mg/kg. Zincol-2016 was grown with zinc fertilizer in a region close to Peshawar (25 km from the study communities), on a combined growing area of 100 ha for use in this effectiveness study and was supported by our collaborating organisation, Fauji Fertilizer Company (FFC) Ltd., (Rawalpindi, Pakistan) throughout this period. Based on the initial sampled soil testing for mineral and organic content, each farmer was provided with the required quantity of seeds and fertilizers (soil and foliar) to optimise zinc supply to the growing crop. These farmers were also advised from time to time on the standard agronomic practices to be followed. The grains were sown between October and November 2019. During the growth, zinc foliar spray (0.1% elemental zinc; 303 g of ZnSO_4_·H_2_O in 100 L of water used over 0.4 ha of crop) was applied at the booting stage, heading stage and a week after the emergence of heads. This crop was harvested using both mechanical and manual methods in the month of May 2020, after an assessment for readiness performed by regional agronomists from FFC. The grains were then packed in the sacks and transported to the project site. Samples of grain from each farm were sent to the University of Nottingham (UoN), UK, for mineral content analysis.

Galaxy variety, which is a standard wheat variety used in Pakistan, was used as the control grain. It was purchased from commercial farms in Punjab province and sent to the project site packed in the sacks identical to those used for the Zincol-2016 variety.

These grains were milled into flour at a local commercial mill and distributed to the families of enrolled participants of the RCT as outlined in the above sections.

### 2.4. Mineral Analysis of Grain and Flour

After the harvest in May 2020, samples of grain were from each farm growing Zincol-2016 wheat variety to assess whether the target zinc content of 40 mg/kg had been achieved prior to the start of the RCT. Control grain, purchased from commercial suppliers, was not analysed as reference data as this variety was already available [12]. Zincol-2016 grain zinc concentration was measured using previously described methods [20]. In brief, approximately 0.4 g (whole grain) was dried, weighed, and soaked in 8 mL 70% Primar Plus™ HNO_3_, at room temperature overnight in 50 mL polypropylene digestion tubes. Samples were hot-block digested (Multicube 48-Anton Parr, PFA-coated graphite hot block, Graz, Austria) for 2 h at 115 °C. Whole-grain zinc and other mineral contents were simultaneously determined by inductively coupled plasma-mass spectrometry (ICP-MS; Thermo Fisher Scientific iCAPQ, Thermo Fisher Scientific, Bremen, Germany).

The flour resulting from each batch of Galaxy and Zincol-2015 grain sent fortnightly for milling, was sampled (*n* = 3–13 flour samples per batch for Galaxy; *n* = 3–6 flour samples each batch for Zincol) from randomly selected sacks for mineral analysis at UoN. Flour samples (0.4 g) were digested as described above and analysed by ICP-MS.

### 2.5. Field Procedures

#### Participant Characteristics, Blood Sample and Diet Data Collection Procedures

Characteristics of the participants, including age, education level achieved, indicators of socioeconomic status, demographics of the household, living conditions (such as water source, kitchen, and toilet facilities), details related to wheat flour consumption and purchase practices, were collected at enrolment (T1) using an interviewer-administered questionnaire.

Blood samples were collected at all five time points of the study to monitor general haematological parameters, plasma zinc concentration, and other mineral biomarkers. The samples were collected by a trained phlebotomist at a health centre for area A. In area B, there was no health centre available; therefore, a temporary clinic was set up in a building adjacent to a school operated by our collaborating organization, Abaseen Foundation Pakistan (AFPK). Whole blood (non-fasting) was drawn from the antecubital vein through a butterfly needle into three types of vacutainers procured from Cytomark^®^, Buckingham, UK (1) tubes with ethylenediaminetetraacetic acid (EDTA) anticoagulant for estimation of routine haematological parameters (2) in trace-element-free tubes containing EDTA anticoagulant to process further for separating plasma (3) in a tube with gel to separate serum. Following the blood drawings, the participants were provided with fruit juice.

A total of three dietary recalls, one each at T3, T4 and T5, using 24 h recall method were carried out, which is sufficient to estimate the nutrient intakes of individuals as well as the proportion at risk of inadequate intakes [21]. Dietary recalls were conducted by trained nutritionists during home visits using a multiple pass method and portion sizes were estimated employing local household measures. The quantity of bread consumed by each participant was estimated from the dietary recalls, and local bread recipes were used to determine daily flour consumption.

### 2.6. Lab Procedures

#### 2.6.1. Haematology

Approximately 20 μL of EDTA whole blood was used to determine complete blood count, including red blood cell count, haemoglobin, mean corpuscular volume (MCV), haematocrit (HCT), and mean corpuscular haemoglobin concentration (MCHC) via an automated portable haematological analyser (Mindray, Shenzhen, China at a local private diagnostic centre (Medicaid Diagnostic Services, Peshawar, Pakistan).

#### 2.6.2. Mineral Analysis

Blood samples were centrifuged to remove plasma within 40 min of sample collection and stored at −80 °C at Khyber Medical University until shipped on dry ice to the UoN. Zinc and other mineral concentrations in the plasma were measured simultaneously using inductively coupled plasma-mass spectrometry as described previously [8]. Seronorm Trace Elements Serum L-1 and L-2 (Nycomed Pharma AS, Billingstad, Norway) were used as the certified reference materials to verify accuracy. Further details are provided in Table A1 and Table A2 of the Appendix B.

#### 2.6.3. Inflammatory Markers and Iron Status

Similar to blood plasma, serum was stored at −80 °C until analysed at Rehman Medical Institute (RMI), Pakistan. Biomarkers of iron status, namely serum transferrin receptor (sTFR) and ferritin (SF), were assessed. sTFR was measured by a particle-enhanced immunoturbidimetric assay (Tina quant sTfR, Roche Diagnostics) on a fully mechanised analyser Cobas 6000 (Roche Diagnostics). Ferritin was quantified on an automated analyser (Abbott Architect ci8200, Abbott Laboratories, Abbott Park, IL, USA), employing a chemiluminescent microparticle immunoassay technology-based commercial kit (Architect ferritin 7K59, Abbott Laboratories).

Alpha 1-acid glycoprotein (AGP) and C-reactive protein (CRP), the two markers recommended for adjusting plasma zinc and serum ferritin, were measured using commercial kits and Abbott Architect ci8200 automated analyser (Abbott, Abbott Park, IL, USA). Quantia A-1- alpha 1-AGP and Multigen CRP Vario assay kits (from Abbott Laboratories) were used for estimating AGP and CRP, respectively, by an immunoturbidimetric method.

These assays were performed according to the manufacturer’s instructions including suggested calibrators and controls. Laboratory-quality control performance was tracked using the Westgard rule criteria on the Levey–Jennings chart [22].

#### 2.6.4. Adjustments for Inflammation and Cut-Offs for Defining Deficiencies

Decile analysis and correlations between plasma zinc and inflammatory markers were performed to understand any requisites for adjusting plasma zinc concentration for inflammation based on recent guidelines from the International Zinc Nutrition Consultative Group [23].

Ferritin levels were adjusted for inflammation-related high iron stores as per the World Health Organization (WHO) recommendations [24] using arithmetic correction factor approach as proposed by Thurnham et al. [25], by grouping into the four inflammation groups: (1) reference (both CRP concentration ≤ 5 mg/L and AGP concentration ≤ 1 g/L); (2) incubation (CRP concentration > 5 mg/L and AGP concentration ≤ 1 g/L); (3) early convalescence (both CRP concentration > 5 mg/L and AGP concentration > 1 g/L); and (4) late convalescence (CRP concentration ≤ 5 mg/L and AGP concentration > 1 g/L). Correction factors were derived by a ratio of geometric means of the reference group to those of the respective inflammation group. These factors: 1.0, 0.92, 0.58, 0.72 were applied to the above four groups, respectively.

Zinc deficiency was defined as PZC levels < 650 µg/L for girls below 10 years of age and <660 µg/L for girls 10 years or older, according to IZiNCG recommendations for morning non-fasting state [5]. Low plasma iron was defined as 10.7 µmoles/L [26]. A serum ferritin (SF) concentration of <15 ng/mL [24] and TFR > 4.59 mg/L [27] were considered as storage iron deficiency and functional iron deficiency, respectively. Iron deficiency was defined by either SF < 15 ng/mL or sTFR > 4.59 mg/L.

Cut-offs for Hb levels as per WHO criteria were used for determining anaemia and its grade [28]. Participants were considered anaemic if Hb levels were <11.5 g/dL for girls aged below 12 years and <12.0 g/dL for those aged 12 years and above. They were considered severely or moderately anaemic if Hb < 8.0 g/dL or ranged between 8.0–10.9 g/dL, respectively. Girls were considered to have mild anaemia if their Hb ranged between 11.0–11.4 g/dL (<12 years) and 11.0–11.9 g/dL (12 years and above). Iron deficiency anaemia was defined if both anaemia and iron deficiency was present according to definitions above.

CRP levels >0.5 mg/dL or AGP levels > 100 mg/dL were used to identify inflammation [5,25]. Age dependent cut-offs for plasma copper were used to identify copper deficiency [29]: <750 μg/L for girls below 10 years; <640 μg/L for 10–12.5 years; and <570 μg/L for those over 12.5 years in age. Plasma selenium concentrations <41.8 μg/L reflected selenium deficiency [30]. Based on the reference range provided by Centers for Disease Control and Prevention (CDC) for 6–18-year-old children, a value less below 33.5% for HCT; 74.7 fL for MCV; 32.3 g/dL MCHC; and 3.84 × 10^6^/µL for RBC counts were used to define low levels of these haematological markers [31].

### 2.7. Data Analysis

Data on the outcome variables collected at T3 (beginning of phase II/end of phase I) were considered as baseline for the intervention phase. Outcome measures relate to participants rather than clusters. Effect of intervention was assessed at the midpoint (T4) and endline (T5) of the study for all continuous variables, and comparisons between the two arms at all three time points (T3, T4, T5) were made for binary and categorical variables. The primary outcome variable was unadjusted PZC. No adjustments for inflammation were required as the results of the correlation and decile analysis did not indicate any association between plasma zinc concentrations and CRP and/or AGP (Appendix A Appendix A). Secondary outcomes variables were: (i) those related to blood biochemistry such as plasma minerals (plasma iron, selenium, copper), iron status markers (serum Ferritin and sTFR), inflammatory markers (serum AGP/CRP), haematological measures (Haemoglobin, RBC count, MCV, MCHC, HCT); and (ii) dietary intakes of nutrients (zinc and iron).

For outcomes pertaining to blood biomarkers, both primary and secondary continuous variables at midpoint and endline were analysed in the context of linear mixed-effects models, which included random cluster effects to account for the cluster-based nature of the randomisation process. In addition to the study groups, the models included the baseline value (T3) as a continuous covariate. We undertook these analyses on an intention-to-treat basis and adopted the restricted maximum-likelihood method. For linear models, linear regression coefficients (β) and 95% confidence intervals (CI) are reported.

Pearson chi-squared (*X*^2^) tests of independence were used at each of the experimental timepoints to undertake bivariate cross-tabulation comparisons to test for differences in the number of participants exhibiting binary or categorical outcomes (such as the prevalence of mineral deficiencies, various grades of anemia) between trial arms. Probability values for chi-squared analyses were calculated by Monte Carlo simulation. Statistical significance for all analyses was accepted as the *p* < 0.05 level, with the use of IBM SPSS software, version 27.0, IBM Corp. (Armonk, NY, USA).

## 3. Results

Eligible households (*n* = 674) were approached from 34 clusters, and a total of 517 adolescents from 486 households were recruited to participate in the study, which initiated in November 2019 and ended in March 2021. A total of 110 participants dropped out of the study for reasons given as: migration/no-show during follow-ups (*n* = 22); withdrew consent because of certain constraints (related to religion/culture/health) or an unspecified cause (*n* = 70); had to be excluded as adolescents got married and moved out of the enrolled clusters in the study and therefore data/sample collection was not feasible (*n* = 18). The CONSORT flow diagram is presented in Figure 2. The overall participant retention was 79%. The distribution of participant dropout at various study points was independent of the study arm allocation (x^2^_(4)_ = 9.336; *p* = 0.053).

Compliance with the study treatment was high. Household visits with inbuilt spot checks (*n* = 1953 total visits for 399 households) indicated that the study flour was consumed regularly (*n* = 1923) while the data on the current consumption was missing (*n* = 30) for a few household visits. Of the total visits for which a response was received on the quality of the project flour (*n* = 1934 response present; *n* = 19 missing data), the study flour was found to be reported consistently better (*n* = 1850) or the same as the regular flour (*n* = 83) which the households were purchasing before the commencement of the trial. It was only on one occasion that the quality of the study flour was reported to be worse than the regular flour previously used by the family.

### 3.1. Participant Characteristics

The general characteristics of the participating households and participant characteristics are summarized in Table 1. The average age of girls at enrolment (*n* = 517) was 12.1 ± 1.7 years, with a range of 8.6–15.3 years. More than half of the participants (56.4%) were not attending school at the time of the study; 30.3% had never attended school; 22.5% had dropped out of school; 3.7% were missing data). Of those who had ever been to school (66.1%), most had only partially achieved primary education (36.1%) at the time of enrolment to the study. Nearly half (46.1%) of the girls had attained menarche at the time of enrolment.

The family size of participating households varied from 4–36 members with a mean of 10.5 ± 4.8. Migration of families out of the study area was uncommon and the majority of 85.3% had resided in the present location for more than 10 years. The source of income for most households was a daily wage through labour in at a brick kiln (59.1%), followed by employment in either private (19.1%) or government (10.9%) sector, and only 1.4% were generating income through farming as an occupation. For the majority, household income was below 20,000 PKR (61.2%). About half the homes were solely built of mud and straw, known as *katcha* (46.8%), while the rest were either *pakka* (cemented) or a mix of both types, and all homes comprised three rooms on average. Toilet facilities were present in nearly all homes (91.4%). Overall, 81.3% of the respondents reported being the homeowner, and 14.8% were renting. Close to 90% of the households possessed an alternative source of energy, such as solar panels or an uninterruptible power supply device, and over a quarter owned a fridge/freezer (28.6%).

Most households had a separate designated structure for a kitchen (79.3%), which may (66.3%) or may not (13%) have a roof located within the household compound. Some cooked in an open space with temporary arrangements on a daily basis (19.1%) and under 2% prepared meals inside the room they were living in. The source of drinking water was a local borehole with either an electric motor (90.5%) or a handpump (2.7%) for drawing water. Engagement with the national vaccination program for young children was almost universal among households (95.9%). Morbidity was common: 40% reported a diarrhoeal incidence among at least one child (<5 years) in the past month; 40% reported respiratory tract infection (RTI) episode among at least one child (<5 years) in the past month; and 22% reported that at least one adolescent girl had an RTI in the past month. Household flour was most frequently purchased from the local market, except for a small percentage (8.6%) of households who also grew their own wheat. Households reported consuming between 10 to 300 kg of flour per month, with an average of 107.4 kg per month. Monthly consumption of other staples, namely maize and rice, averaged 6.7± 8.2 kg. Payment by cash and/or credit for wheat flour was common practice. When asked about their preference for receiving flour during the first phase of the study, the majority (65.1%) preferred receiving the flour directly over cash or coupon to obtain flour through local suppliers.

Some blood samples were lost to individual outcome measure analyses due to the non-viability of the sample for various reasons, including haemolysis or low sample volume, missing (did not turn up for sampling of blood) and extreme values. The number of samples analysed for each outcome measure are provided in the corresponding results section, either in text or tables. Baseline values (at T3) for all haematological outcome measures are presented in Table 2. Mean PZC was 624.7 ± 88.2 µg/L, and the prevalence of zinc deficiency amongst the adolescent girls was remarkably high (68.8%). Iron deficiency, assessed using SF and sTFR indicators, affected 40% of the adolescent girls, and 9.3% exhibited its functional effects in the form of iron deficiency anaemia. At baseline, there was no significant difference between the two arms in the primary outcome measure (PZC). All other biochemical markers were similar for both study arms, except for the prevalence of depleted iron stores (SF < 15 µg/L), the prevalence of copper deficiency and plasma selenium concentration. The prevalence of depleted iron stores was significantly greater (*p* = 0.046) in the intervention arm at baseline (40.3%) compared to the control arm (30.8%) and the prevalence of copper deficiency was significantly higher in the control arm (0.5% intervention arm vs. 3.2% control arm; *p* = 0.042). Mean plasma selenium concentration was significantly lower (*p* = 0.020) in the intervention arm (101.6 ± 12.4) compared to the control arm (104.2 ± 14.7); however, none of the participants were selenium deficient (plasma selenium < 41.8 µg/L).

### 3.2. Grain and Flour Analysis

The mean zinc content of the locally grown Zincol-2016 wheat grain was 45.3 ± 10.7 mg/kg and varied extensively from 24.3 to 76.3 mg/kg (*n* = 89). Similarly, the iron content was variable, ranging from 19.6 and 60.5 mg/kg with a mean of 32.3 ± 6.3 mg/kg (*n* = 89).

A total of 75 samples of control (Galaxy) flour and 62 samples of biofortified (Zincol-2016) flour, which were distributed to the households during the intervention phase, were analysed for mineral content. Two samples from the control had to be excluded because of extreme outlier values (outside of 3 × IQR) for zinc and/or other minerals (Fe, Cu, Se, P, Ca). The biofortified flour had statistically significant greater zinc concentration than the control flour (biofortified mean zinc concentration 20.7 ± 5.6 mg/kg vs. control mean zinc concentration 17.0 ± 2.6 mg/kg, *p* < 0.001).

The biofortified flour also had a significantly higher concentration of iron, selenium, phosphorus and calcium compared to the control flour (*p* < 0.01). The copper concentrations of both types of flour were similar (*p* = 0.063). A comparative summary of the mineral content for the two varieties of flour is presented in Table 3.

### 3.3. Impact of Intervention

#### 3.3.1. Contribution to Daily Zinc Intake from Wheat Flour

In Pakistan, wheat flour is consumed primarily in the form of bread, which is part of every meal during the day. The mean consumption of bread by adolescent girls, for whom at least two 24 h dietary recalls were available (*n* = 412), was estimated to be 541 ± 134 g/day (Table 4). The flour provided to the families was processed or extracted to the extent of about 80% based on the preference for “mixed flour” reported by the majority of families during the enrolment survey. By use of a conversion factor of 0.75, derived from the standard recipes of various bread consumption in these study communities, this intake of bread corresponded to a wheat flour consumption of 405 g/day. Accounting for the analysed mineral content of the two varieties of wheat flour provided in the study, this translates to an expected zinc intake of about 6.9 mg/day from control flour and a zinc intake of 8.4 mg/d from biofortified flour, representing a daily dietary increase of 1.5 mg zinc (95% CI:1.46, 1.54) for the intervention group over control. A contribution of an additional 1.2 mg/day (95% CI: 1.18, 1.24) iron intake was expected from the biofortified flour at the mentioned level of flour consumption. No adverse effects due to the consumption of either the control or biofortified flours were reported during the study.

#### 3.3.2. Plasma Zinc Concentration and Prevalence of Zinc Deficiency

The treatment effect was not significant for PZC either at midpoint (β = 5.44; 95% CI: −13.96, 24.85; t = 0.57; *p* = 0.51) or endline of the study (β = 8.58; 95% CI: −14.53, 31.69; t = 0.76; *p* = 0.46) after adjusting for the baseline concentrations (Figure 3). Similarly, no effect of treatment was seen in the prevalence of zinc deficiency at either of the assessment points i.e., TP 4 (X^2^ _(1)_ = 0.08, *p* = 0.78) or TP 5 (X^2^ _(1)_ = 0.71, *p* = 0.40), although an overall reduction in zinc deficiency rates over time was observed in both arms from T3 to T5 (Table 5).

A subgroup analysis of participants with a baseline plasma zinc concentration below the cut-off value indicating deficiency (*n* = 146), did not indicate any differences between the PZC at either later assessment points (*p* = 0.81 at T4; *p* = 0.47 at T5) as presented in Table 5.

#### 3.3.3. Haematology and Adverse Effect

There were no significant differences between trial arms at the midpoint or endline of the study for haemoglobin, haematocrit, MCV and MCHC. In addition, the chi-squared analysis did not reveal any significant differences in the prevalence of anaemia, nor in the percentage of participants with values below the recommended cut-offs (or reference ranges) for the above haematological parameters. Linear regression coefficients (β with 95% CI) for the above continuous outcome variables and the X^2^ values for dichotomous variables (such as the percentage of participants falling below the normal ranges), are presented along with the data summary and *p*-values in the Appendix A. Further, the prevalence of three grades of anaemia, i.e., mild, moderate, and severe, were also independent of the study arm allocation at all three assessment points (Appendix A).

#### 3.3.4. Iron Status

The effect of the intervention on iron status biomarkers is shown in Table 6. No significant differences between trial arms were observed at midpoint or endline assessments for plasma levels of iron (T4, *p* = 0.39; T5 *p* = 0.37), SF adjusted (T4, *p* = 0.14; T5, *p* = 0.17) and sTFR (T4, *p* = 0.97; *p* = 0.44) after adjusting for baseline values. Linear regression coefficients (β) with 95% CI for these outcomes are given in Table 6 together with data summary for plasma/serum levels of the indicators. Chi-squared analysis showed no significant difference between study arms with respect to storage iron deficiency (*X*^2^ _(1)_ = 0.389, *p* = 0.533 at T4; *X*^2^ _(1)_ = 0.063, *p* = 0.802 at T5), or functional iron deficiency (*X*^2^ _(1)_ = 0.544, *p* = 0.461 at T4; *X*^2^ _(1)_ = 1.777, *p* = 0.183 at T5). Iron deficiency (*X*^2^ _(1)_ = 0.046, *p* = 0.829 at T4; *X*^2^ _(1)_ = 0.839, *p* = 0.360 at T5) and iron deficiency anaemia (*X*^2^ _(1)_ = 0.110, *p* = 0.740 at T4; *X*^2^ _(1)_ = 0.592, *p* = 0.442 at T5) were not significantly different between arms.

At T3 (baseline) it was observed that the prevalence of storage iron deficiency (SF > 15 ug/L) was significantly greater (*p* = 0.042) in the intervention group (40.3%) compared to control (30.8%). This difference was reduced at T4 (midpoint) by a greater fall in the prevalence of iron deficiency in the intervention group than in the control group (14.5% and 7.7%, respectively) such that prevalence at the midpoint was 23.1% and 25.8% for the control and intervention arms, respectively. At the T5 (endline) assessment, the prevalence had increased in both groups, but remained comparable (control, 41.4%; intervention, 42.6%). Overall, an increase in the prevalence of storage iron deficiency in the control arm was found to be 11.8% vs. 1.0% in the intervention arm between the baseline and endline of the study.

#### 3.3.5. Other Mineral (Selenium and Copper) Status and Inflammatory Markers

There were no significant differences between control and intervention arms at either T4 (β = 16.882; 95% CI: −7.130, 40.894, t = 1.447; *p* = 0.160) or T5 (β = 18.562; 95% CI: −10.056, 47.180; t = 1.326; *p* = 0.195) for serum copper concentration. Similarly, for serum selenium, there were no significant differences between trial arms at either T4 (β = 0.960; 95% CI: −2.637, 4.556; t = 0.536; *p* = 0.595) or T5 (β = −0.270; 95% CI: −3.483, 2.943; t = −0.170; *p* = 0.866). Prevalence of copper deficiency was low and ranged between 0.7 to 2.9% at any time point. Overall, an increase in prevalence of copper deficiency was 1.5% in the control arm compared to 0.6% in the intervention arm between baseline and endline. For serum copper at T5, there was a significantly greater (*X*^2^ _(1)_ = 4.64, *p* = 0.035) deficiency exhibited in the control arm compared to the intervention arm. However, at the baseline, the rates for copper deficiency in the control arm (3.2%) were also significantly (*p* = 0.046) higher compared to intervention arm (0.5%). None of the participants exhibited any deficiency of selenium throughout the study (plasma levels < 41.8 µg/L).

The mean Cu:Zn ratio, a suggested index with a potential for detecting zinc deficiency, remained in a narrow range between 1.4 to 1.6 throughout the study. The intervention did not have a significant impact on the Cu:Zn ratio at midpoint (β = −0.032; 95% CI: −0.026, 0.090; t = 1.116, p = 0.273) or endline (β = −0.024; 95% CI: −0.032, 0.081; t = 0.876; *p* = 0.388) after adjusting the baseline values.

There was no evidence of a significant difference between the arms in terms of CRP levels and the prevalence of inflammation based on CRP or AGP levels. AGP levels were higher at T4 in the control arm compared to the intervention arm, which reached statistical significance at T5 (β = −4.548; 95% CI: −8.921,−0.175; t = −2.148; *p* = 0.042).

Plasma/serum levels of the above biomarkers and linear regression coefficients (β) with 95% CI as well as prevalence rates, values for the chi-squared test of independence of arms for prevalence rates along with significance values are reported in Table 7.

On exploring the relationship between inflammatory markers, we found that though statistically significant (CRP and selenium; CRP and copper; AGP and copper; *p* < 0.01), these associations (r_s_ = 0.62–0.252) were weak. Spearman correlation coefficient along with *p*-values are provided in Table 8.

## 4. Discussion

Our study highlights the magnitude of zinc deficiency rates among adolescent girls in a low-resource community setting in northwest Pakistan. Based on baseline PZC, 68.8% of adolescent girls were found to be zinc-deficient, which is more than double the rates previously observed for women of reproductive age (WRA) by us in the same community [7,14] and higher than reported in other studies. The recent national nutrition survey (NNS) of Pakistan reported that the highest prevalence of zinc deficiency was observed in rural Punjab, where 27.6% of WRA, regardless of pregnancy status, were zinc-deficient [7]. A secondary analysis of 15–19-year-old Pakistani girls estimated 54% to be anaemic, 42% zinc-deficient and 21.4% to have iron deficiency anaemia [9]. These discrepancies could be attributed to a lower cut-off used for estimating zinc deficiency in the national nutrition survey (<60 µg/dL), or due to the differences in age groups compared. In our present study, two of the participants were less than 10 years at the time baseline assessment for outcome measures were performed. We have used the age-specific cut-offs of 65 µg/dL and 66 µg/dL for girls below 10 years and above 10 years, respectively, under morning non-fasting conditions as suggested by IZiNCG [5]. Since we previously found a 30% prevalence of zinc deficiency in non-pregnant, non-lactating women (using the recommended cut-off of 66 µg/dL for this population) from the same community, our observations suggest that interventions targeting improvement of zinc status should be initiated in the early phase of life for females. This becomes imperative in Pakistan given that early marriages and childbearing is common in rural communities where the majority of adolescents reside (62.9%), and particularly in the KP region where marriage before the age of 18 years of age (28.8% for KP and 33.1% for KP-NMD), and childbearing among 15–19-year-olds (11.9% KP and 19.3% KP-NMD) are the highest in the country according to the most recent national survey [7]. In addition, the mean plasma Cu:Zn ratio, another putative indicator of zinc status, was found to be close to 1.5 at any point of the study (Cu:Zn ratio of 0.7–1.0 is optimal) and consistent with our findings in WRA from the same community [12]. The high prevalence of zinc deficiency clearly emphasizes the need to improve zinc nutrition in this setting. Biofortification of staple crops has the potential of being a cost-effective approach to improve micronutrient intakes in LMICs. To the best of our knowledge, BiZIFED2 is the first large cluster randomised controlled trial investigating the effectiveness of zinc biofortified flour consumption on micronutrient status among adolescent girls, and in Pakistan.

In the present study, the high-zinc variety ‘Zincol-2016′ wheat used in the RCT was grown with zinc fertilizers by the local farmers and met the target grain zinc concentration of >40 mg/kg. Although technical support was provided for fertilizer application, the crop was grown under “real world” conditions, in contrast to highly controlled experimental conditions, and thus indicative of the crop performance during future scale-up initiatives with similar levels of support. The estimated average daily consumption of flour by the study participants was 405 g per person. This consumption is slightly more than the average per capita national consumption (340 g/day) of wheat flour. This intake is plausible given that the community subsists predominantly on a wheat-based diet and, as the wheat flour was provided free of charge for the entire study period, it is likely to have replaced, at least to some extent, other cereals in the usual diet. Despite an estimated additional mean intake of 1.5 ± 0.40 mg/day from biofortified flour, which corresponded to a 22% increase in daily zinc intake from the non-biofortified control flour, regular consumption of biofortified flour for 25 weeks did not have any significant effect on PZC or the prevalence of zinc deficiency based on PZC among the participants. 

Although it can be argued that the contribution of additional absorbed zinc from the biofortified flour may be low because of the presence of high phytate intakes from wheat bread among the study participants to reflect any meaningful changes in the PZC, the limitations of PCZ as a marker are well-known and cannot be ruled out. Our finding is not surprising given that PZC is not sensitive to respond to moderate changes in zinc intake, especially in the form of food as compared to supplements. The lack of a robust, sensitive biomarker for zinc status creates a challenge when attempting to assess the impact of biofortification [33,34]. Our findings lack an intervention effect on PZC and are consistent with several previous studies [33,35,36]. A recent RCT evaluating the efficacy of feeding zinc biofortified rice to Bangladeshi children for 9 months explored the feasibly of using an emerging marker of zinc, fatty acid desaturase (FADS) activity, in addition to PZC [35]. The authors found no significant effect of the intervention on PZC, the prevalence of zinc deficiency, or FADS activity, which they suggested may be attributed to low additional zinc intake (1 mg/day). The group highlighted their success in accurately measuring this potential biomarker in a low resource setting, thus indicating the suitability of the method in field studies. Zyba et al. found that a moderate 4 mg/day increase in dietary zinc among men aged 18–45 years, an amount similar to what would be expected from zinc-biofortified crops, improved zinc absorption and DNA fragmentation but not PZC [37]. Further identification and testing of such biomarkers are needed to establish a sensitive and robust marker for understanding the effect of food-based interventions in the context of zinc. In our study, other PZC dependent indicators of zinc status, such as rates of low PZC levels and the Cu:Zn ratio, remained comparable for both arms throughout the study duration.

Analysis of the blood samples for other measures revealed almost a third of the participants to be iron deficient based on serum ferritin levels adjusted for inflammation. Although this is commensurate with national rates, it is at least 40% higher than that reported for the KP region in the recent NNS of Pakistan [7]. Anaemia does not appear to be a major problem among adolescent girls in this community. Concerns around selenium deficiency in northeast Pakistan have been raised by Ahmad et al. based on daily selenium intake estimations from urinary selenium concentration in varied population subgroups (*n* = 451) [38]. However, based on plasma selenium concentrations, we did not find deficiencies among our study participants at any point during the study.

It is interesting to note that at baseline, there was a significantly greater prevalence of storage iron (ferritin) deficiency in the intervention arm compared with the control arm. This difference was resolved at the midpoint and remained so at the endline. An explanation for this could be the higher concentration of minerals such as iron in the biofortified wheat compared to the control, which could have positively impacted the ferritin. Studies have shown that iron (and zinc) biofortified crops improve total absorbed iron and ferritin status [39,40,41,42,43]. However, an unexplained sudden dip in the prevalence of storage iron deficiency at the midpoint for both arms limit us from deriving any definitive conclusions. The overall prevalence of copper deficiency at baseline was very low (baseline 1.7%, midpoint 0.7%, endpoint 3.0%), but the intervention did have a significant positive impact compared to control based on achieving a threshold plasma copper concentration of < 640 μg/L for 10–12.5 years; and <570 μg/L for those over 12.5 years in age. Interpretation of this observation is speculative, but since the majority of plasma copper is bound to ceruloplasmin which is an acute phase protein, the difference observed may be related to differences in the presence of inflammation in the two groups.

The prevalence of inflammation in our study participants, based on serum AGP and CRP concentrations, was low and similarly distributed between the two arms of the study at any given assessment time point. We found a significantly higher AGP concentration (a systemic marker of inflammation) at the end of the study in the control arm compared to the intervention arm. Zinc is a negative acute-phase reactant; hence its concentration decreases in the presence of inflammation [44], which has implications for accurate estimates of population-level zinc status. While several analyses have attempted to understand the effect of AGP on plasma zinc levels to draw consensus on how to control for the effect of inflammation on serum zinc, the effect of dietary zinc intake on AGP levels is not well studied in humans [45,46]. A subgroup analysis of Tanzanian infants participating in a randomised, double-blind, placebo-controlled trial who received a daily, oral supplementation of zinc or multivitamins, a combination of the two, or placebo for 18 months starting at 6 weeks of age, failed to observe any significant treatment effects of zinc or multivitamins on systemic inflammation including serum AGP concentration [47]. While a systematic review and metanalysis of RCTs to understand the effect of zinc supplementation on the CRP concluded that zinc is beneficial in reducing the serum concentration of this inflammatory biomarker, particularly at high doses [48], we are the first to report a positive effect of a moderate increase in zinc intake from food on AGP. Inflammatory biomarkers are useful indicators of infection in health care settings and have great value for monitoring chronic disease activity and overall health status in the wider population. Considerable research has shown that inflammatory biomarkers can predict mortality in adults with chronic conditions such as type 2 diabetes and cardiovascular diseases [49]. AGP has been identified as one of the strongest predictors of all-cause mortality [50] and systemic inflammation biomarkers have been linked to poor child growth [51]. As zinc plays an important role in cell-mediated immunity and is an antioxidant and anti-inflammatory agent, its deficiency is also associated with non-communicable diseases (NCDs). A recent meta-analysis of zinc supplementation trials found that the supplementation significantly improved risk factors for both cardiovascular diseases and type 2 diabetes mellitus compared to placebo [52]. It would be worthwhile to further explore if such benefits can be realized from zinc biofortification to address the dual burden of malnutrition.

The strengths of our study include: (1) a large sample size, double-blind, randomised cluster-controlled design; (2) a considerable stabilization period after the introduction of free of charge flour; (3) successful local production and milling of the Zincol-2016 grain demonstrating the possibility of scaling up; (4) provision of the flour for the entire household ensuring that family meals using the flour were consumed by the adolescent girls as a part of their usual family meals; (5) high compliance to the study flour consumption by the participating families. One main limitation of the study was the modest differential zinc content between the intervention and control flour (3.3 mg/kg). This was markedly lower than the concentrations reported in a study conducted in India to test the efficacy of agronomic biofortification in children and WRA [36]. In the referred study, the zinc content was 30 mg/kg for the biofortified variety, while the low zinc variety flour had 20 mg/kg, providing a differential of 10 mg/kg. It is also slightly below our expectations based on our previous estimates of the differential zinc content of biofortified vs. control flour from our foundation BIZIFED efficacy study, where both the Zincol-2016 and the Galaxy wheat were grown under carefully controlled conditions, closely monitored by our study collaborator (MHZ), and fertilizer application to the Zincol-2016 grain only [12]. In this previous study, Zincol-2016 grain achieved a zinc concentration of 49.3 mg/kg compared to 22.2 mg/Kg for Galaxy grain. Based on an estimated 50% reduction in zinc concentration post-milling for white flour, this translated to estimated white flour zinc concentrations of 24.5 and 11.1 mg/kg for biofortified and control flour, respectively.

In our present study, the mineral analysis of Zincol-2016 grain from each of the participating farms was conducted soon after the harvest to confirm that the mean zinc content had met the target of >40 mg/kg. The mean grain zinc concentration was 45.3 mg/kg, although it varied from 24.32 to 76.34 mg/kg. The milling process resulted in a yield of approximately 80% after the bran had been removed. Analysis of the flour revealed a 50% reduction in mean zinc concentration, as predicted previously [12]. The galaxy grain was purchased from commercial suppliers and the zinc content was not measured in the present study. The average grain zinc content of standard wheat varieties grown without zinc fertilizer in Pakistan is 29.0 mg/kg [53]. The control flour had a mean zinc concentration of 17.0 mg/kg, which is slightly higher than expected based on the typical zinc content of standard varieties in Pakistan [53], or our previously reported Galaxy grain zinc concentration [12]. This emphasizes the need for a sensitive biomarker of zinc status, as mentioned previously, to delineate clear conclusions on benefits from a moderate gain in dietary zinc intake through interventions such as biofortification. DNA fragmentation and fatty acid ratio analyses (determining FADS activity) are currently underway for a subgroup of our BiZiFED2 study participants to help understand the potential of these emerging biomarkers in the above context.

## 5. Conclusions

In summary, the findings of the present study demonstrate that consumption of locally grown and processed biofortified wheat resulted in a modest (22%) increase in total dietary zinc intake in rural communities where a monotonous diet predominant in staples is prevalent. This increase in the daily zinc intake of 1.5 mg/day for 5.5 months did not translate to an increase in PZC. However, the intervention points toward potential benefits of biofortified wheat flour consumption in terms of improving the prevalence of low iron status among adolescent girls. The additional zinc intake was associated with lower mean serum AGP levels in the study population. This finding opens avenues to understand if any such effect of biofortification exists on systemic inflammation among other population subgroups, particularly those at risk of developing NCDs.

## Figures and Tables

**Figure 1 nutrients-14-01657-f001:**
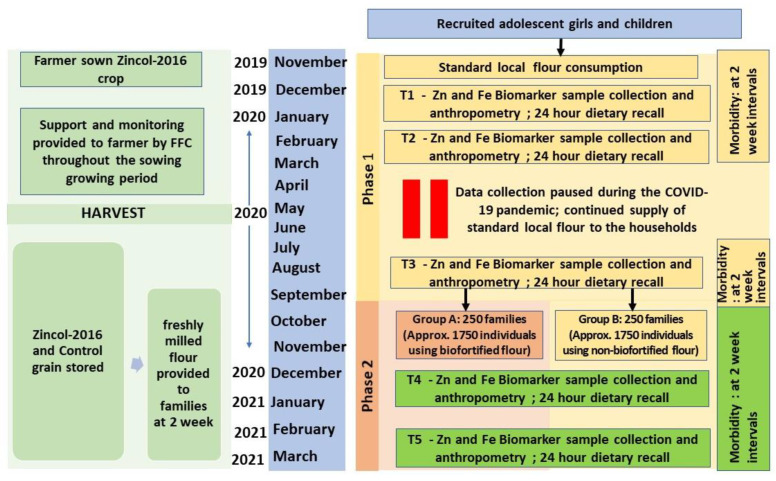
Schematic overview of the study design. FFC, Fauji Fertilizer Company; HH, Households; COVID-19, Coronavirus disease.

**Figure 2 nutrients-14-01657-f002:**
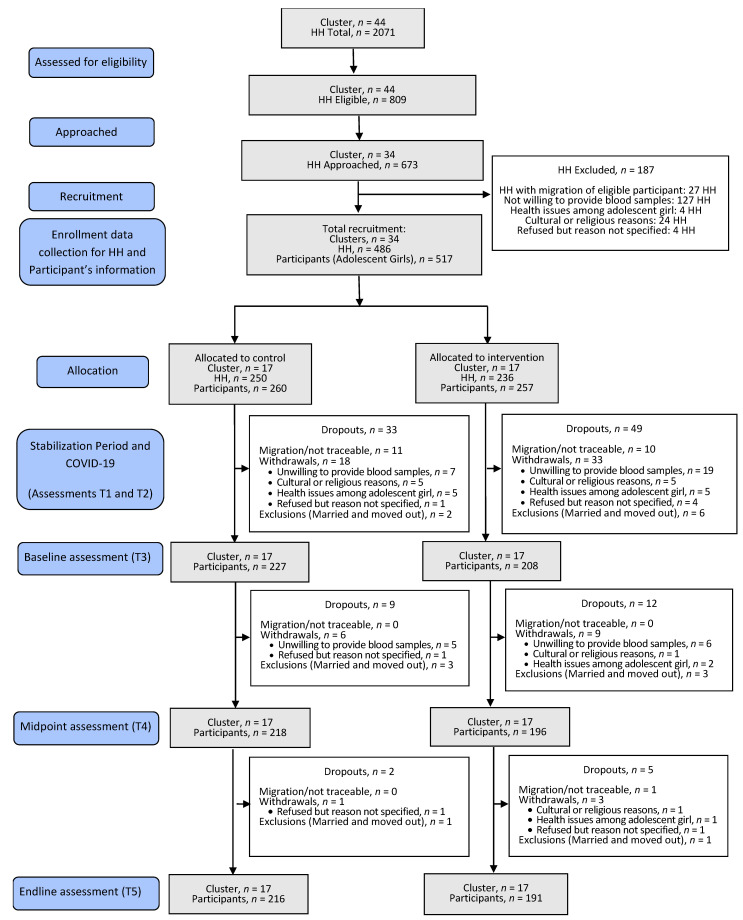
CONSORT flow diagram for the study. CONSORT, Consolidated Standards of Reporting Trials; HH, Households.

**Figure 3 nutrients-14-01657-f003:**
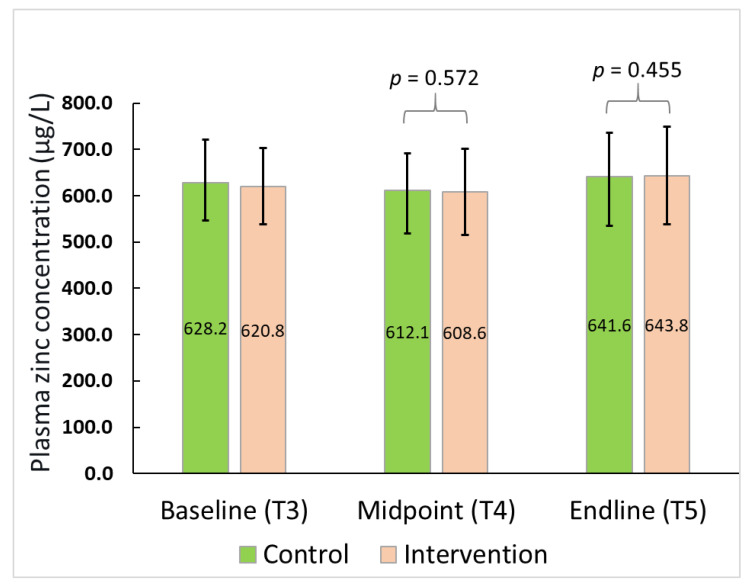
Plasma zinc concentration for the participants over time.

**Table 1 nutrients-14-01657-t001:** Baseline characteristics of the participating households (*n* = 486) and general characteristics of the recruited adolescent girls (*n* = 517) ^a^ by the study arms.

	Total				Arm 1 (Control in Phase 2)				Arm 2 (Intervention in Phase 2)				*p* ^b^
	*n*	Mean (SD)	Median (Range)	*n* (%)	*n*	Mean (SD)	Median (Range)	*n* (%)	*n*	Mean (SD)	Median (Range)	*n* (%)	
**Participant Characteristics:**													
Age	517	12.1 (1.7)	12.0 (8.6–15.3)	-	260	12.2 (1.7)	12.0 (9.3–15.3)		257	12.1 (1.7)	12.1 (8.6–15.3)		0.954
Attending school	512				259				253				
*Yes*				223 (43.6)				104 (40.2)				119 (47.0)	0.116
*No*				289 (56.4)				155 (59.8)				134 (53.0)	
Ever attended school	512				259				253				
*Attending school*				223 (43.6)				104 (40.2)				119 (47.0)	0.106
*School dropout*				115 (22.5)				54 (20.8)				61 (24.1)	
*Never been to school*				155 (30.3)				91 (35.1)				64 (25.3)	
*Unknown (Missing)*				19 (3.7)				10 (3.9)				9 (3.6)	
Education level ^c^	512				259				253				
*Primary*				185 (36.1)				80 (30.9)				105 (41.5)	0.034
*Middle*				65 (12.7)				39 (15.1)				26 (10.3)	
*High*				12 (2.3)				7 (2.7)				5 (2.0)	
*Attending school but level missing*				18 (3.5)				6 (2.3)				12 (4.7)	
*School dropout but level unknown*				58 (11.3)				26 (10.0)				32 (12.6)	
*Never been to school*				155 (30.3)				91 (35.1)				64 (25.3)	
*any info on school/education level missing*				19 (3.7)				10 (3.9)				9 (3.6)	
Menarche attained	506				256				250				0.580
*Yes*				235 (46.4)				122 (47.7)				113 (45.2)	
*No*				271 (53.6)				134 (52.3)				137 (54.8)	
**Household Features:**													
**Household demography ^d^**	486				250				236				
Family Size		10.5 (4.8)	9.0 (4.0–36.0)			10.5 (4.5)	9.0 (4.0–28.0			10.5 (5.0)	9.0 (4.0–36.0)		0.993
No. of young children in HH		2.1 (1.5)	2.0 (1.0–10.0)			2.1 (1.4)	2.0 (1.0–8.0)			2.2 (1.6)	2.0 (1.0–10.0)		0.735
No. of older Children in HH		1.5(1.1)	1.0 (0.0–8.0)			1.5 (1.1)	1.0 (0.0–8.0			1.5 (1.1)	1.0 (0.0–6.0)		0.717
No. of adolescents in HH		3.0(1.6)	3.0 (0.0–11.0)			3.0 (1.6)	3.0 (0.0–10.0)			2.9 (1.7)	3.0 (0.0–11.0)		0.316
No. of adolescent girls in HH		1.7(1.1)	1.0 (0.0–7.0)			1.8 (1.1)	2.0 (0.0–6.0)			1.6 (1.2)	1.0 (0.0–7.0)		0.214
No. of adult males in the HH		1.9(1.4)	1.0 (0.0–8.0)			1.9 (1.4)	1.0 (0.0–8.0)			1.9 (1.4)	1.0 (0.0–8.0)		0.817
No. of adult Females in the HH		2.0(1.3)	2.0 (0.0–7.0)			2.0 (1.3)	2.0 (0.0–6.0)			2.1 (1.3)	2.0 (1.0–7.0)		0.429
**Length of stay in the area**	483				249				234				0.769
*Less than 5 years*				35(7.2)				16 (6.4)				19 (8.1)	
*between 5 to 10 years*				36(7.5)				19 (7.6)				17 (7.3)	
*10 years and above*				412(85.3)				214 (85.9)				198 (84.6)	
**Socio-economic status and living conditions**													
Source of Income ^e^	486				250				236				
*Daily wages*				287 (59.1)				150 (60.0)				137 (58.1)	0.662
*Farming*				7 (1.4)				4 (1.6)				3 (1.3)	0.761
*Business*				21 (4.3)				15 (6.0)				6 (2.5)	0.061
*Govt Job*				53 (10.9)				23 (9.2)				30 (12.7)	0.214
*Private Job*				93 (19.1)				49 (19.6)				44 (18.6)	0.789
*Charity*				1 (0.2)				1 (0.4)				0 (0.0)	0.331
*Pension*				1 (0.2)				1 (0.4)				0 (0.0)	0.331
*Driver*				1 (0.2)				1 (0.4)				0 (0.0)	0.331
*Abroad ^f^*				4 (0.8)				2 (0.8)				2 (0.8)	0.954
*Others (unspecified)*				23(4.7)				10 (4.0)				13 (5.5)	0.434
Monthly Income	485				250				235				0.423
*HH income <20,000 PKR*				297 (61.2)				147 (58.8)				150 (63.8)	
*HH income between 20,000–40,000 PKR*				173 (35.7)				96 (38.4)				77 (32.8)	
*HH income >40,000 PKR*				15 (3.1)				7 (2.8)				8 (3.4)	
Monthly Expenditure	483				248				235				0.477
*HH with expenditure <20,000 PKR*				228 (47.2)				112 (45.2)				116 (49.4)	
*HH with expenditure 20,000–40,000 PKR*				239 (49.5)				129 (52.0)				110 (46.8)	
*HH with expenditure >40,000 PKR*				16 (3.3)				7 (2.8)				9 (3.8)	
Possession of Assets ^e^	454				238				216				
*Refrigerator/Freezer*				130 (28.6)				66 (27.7)				64 (29.6)	0.655
*TV/Cable/Dish*				33 (7.3)				15 (6.3)				18 (8.3)	0.405
*Solar Panels/UPS*				400 (88.1)				209 (87.8)				191 (88.4)	0.841
*Motorcar*				39 (8.6)				19 (8.0)				20 (9.3)	0.628
*Motorcycle*				88 (19.4)				45 (18.9)				43 (19.9)	0.788
*Rikshaw/Chingchi ^g^*				5 (1.1)				1(0.4)				4 (1.9)	0.144
House Ownership	486				250				236				0.740
*Own*				395 (81.3)				200 (80.0)				195 (82.6)	
*Rent*				72 (14.8)				40 (16.0)				32 (13.6)	
*Free tenant*				19 (3.9)				10 (4.0)				9 (3.8)	
Structure of House	485				249				236				0.765
*Katcha (mud and straw)*				227 (46.8)				113 (45.4)				114 (48.3)	
*Pakka (cemented)*				143 (29.5)				74 (29.7)				69 (29.2)	
*Mix of katcha and pakka*				115 (23.7)				62 (24.9)				53 (22.5)	
Number of Rooms house	481	3.1 (1.8)	3.0 (1.0–11.0)		248	3.2 (1.9)	3.0 (1.0–11.0)		233	3.0 (1.7)	3.0 (1.0–11.0)		0.605
Toilet present in House	486				250				236				0.845
*Yes*				444 (91.4)				229 (91.6)				215 (91.1)	
*No*				42 (8.6)				21 (8.4)				21 (8.9)	
No. of toilets in House	466	1.0 (0.6)	1.0 (0.0–4.0)		242	1.1 (0.6)	1.0 (0.0–4.0)		224	1.0 (0.5)	1.0 (0.0–4.0)		0.614
**Food and Water**													
Meal preparation facility in House	486				250				236				0.781
*Purpose-built Kitchen*				9 (1.9)				6 (2.4)				3 (1.3)	
*Separate & covered (separate structure with roof)*				313 (64.4)				163 (65.2)				150 (63.6)	
*Separate & uncovered (separate structure without roof)*				63 (13.0)				31 (12.4)				32 (13.6)	
*within room (inside room living in)*				8 (1.6)				5 (2.0)				3 (1.3)	
*Open space (temporary arrangement)*				93 (19.1)				45 (18.0)				48 (20.3)	
Source of drinking water	484				249				235				0.009
*Borewell*				438 (90.5)				234 (94.0)				204 (86.8)	
*Handpump*				13 (2.7)				2 (0.8)				11 (4.7)	
*Pipeline (spring)*				5 (1.0)				4 (1.6)				1 (0.4)	
*Neighbour (dependent on neighbours)*				14 (2.9)				5 (2.0)				9 (3.8)	
*Others*				14 (2.9)				4 (1.6)				10 (4.3)	
**Health**													
Engagement of HH with the vaccination program	466				243				223				0.006
*Yes*				447 (95.9)				239 (98.4)				208 (93.3)	
*No*				19 (4.1)				4 (1.6)				15 (6.7)	
Diarrheal incidence among children (1–5 years of age) within HH in past month	486				250				236				0.974
*Yes*				196 (40.3)				101 (40.4)				95 (40.3)	
*No*				290 (59.7)				149 (59.6)				141(59.7)	
RTI incidence among children 1–5 years within HH in past month	486				250				236				0.039
*Yes*				198 (40.7)				113 (45.2)				85 (36.0)	
*No*				288 (59.3)				137 (54.8)				151 (64.0)	
RTI incidence among adolescent girls (10–16 years of age) within HH in past month	486				250				236				0.005
*Yes*				107 (22.0)				68 (27.2)				39 (16.5)	
*No*				379 (78.0)				182 (72.8)				197 (83.5)	
* **Flour consumption and purchase** *													
Type of flour used	486				250				236				0.062
*White*				93 (19.1)				47 (18.8)				46 (19.5)	
*Brown*				101(20.8)				59 (23.6)				42 (17.8)	
*Mix*				282 (58.0)				143 (57.2)				139 (58.9)	
*Brown and white*				9 (1.9)				1 (0.4)				8 (3.4)	
*Mix and white*				1(0.2)				0 (0.0)				1 (0.4)	
Source of flour procurement	486				250				236				0.109
*Purchase from market*				445 (91.6)				224 (89.6)				221 (93.6)	
*Self-grown and purchased from market*				41 (8.4)				26 (10.4)				15 (6.4)	
Monthly flour consumption (Kgs)	485	107.4 (45.1)	100.0 (10.0–300.0)		249	107.8 (44.2)	100.0 (40.0–300.0		236	107.1 (46.1)	100.0 (10.0–300.0)		0.882
Monthly consumption of other staple ^h^ (Kgs)	443	6.7 (8.2)	4.5 (0.0–120.0)		230	7.2 (10.1)	4.3 (0.0–120.0)		213	6.2 (5.6)	5.0 (0.0–40.0)		0.280
Purchase practice for flour (mode of payment)	479				248				231				0.678
*Cash*				237 (49.5)				126 (50.8)				111 (48.1)	
*Credit*				168 (35.1)				87 (35.1)				81 (35.1)	
*Cash and Credit*				74 (15.4)				35 (14.1)				39 (16.9)	
Preferred term of supply in the first six months ^i^	481				247				234				0.354
*Flour*				313 (65.1)				168 (68.0)				145 (62.0)	
*Cash*				94 (19.5)				43 (17.4)				51 (21.8)	
*Coupon*				74 (15.4)				36 (14.6)				38 (16.2)	

All data presented in the table were collected at the time of enrolment. HH, household; UPS, device for uninterruptible power supply. ^a^ One or more participants were included from each household. ^b^ *p*-values obtained using linear mixed models adjusted for cluster effect to test differences between the groups for continuous variable and categorical variables by Pearson’s chi-squared test. Significance was set at *p* < 0.05. ^c^ Primary schooling is defined as grade 1–5 (grade 1 to ≤5), middle as grade 6–8 (grade 6 to ≤8), high as grade 9–10 grade (grade 9 to ≤10) irrespective of the girl is attending school at present or had dropped out. ^d^ Population sub-groups defined as young children: ≤5 years of age; older children: ≥5 to <10 years of age; adolescents: ≥10 to ≤19 years of age; Adults: >19 years of age. ^e^ Descriptive indicates number (%) of only those who responded ‘yes’ against a category of source of income/possession of a particular asset. ^f^ Family member(s) working outside of the country as main source of HH income. ^g^ A mode of transportation that are generally battery-driven tricycles and used for carrying goods or people. ^h^ Other cereals include rice and maize, collectively. ^i^ For the duration of phase 1 (stabilization phase).

**Table 2 nutrients-14-01657-t002:** Baseline values for haematological outcome measures of the enrolled adolescent girls.

	*n*	Total	*n*	Control	*n*	Intervention	*p* *
Plasma zinc concentrations, PZC (µg/L)	420	624.7 ± 88.2	221	628.2 ± 93.6	199	620.8 ± 81.7	0.893
Zinc deficiency (PZC < 650 µg/L for age < 10 years or 660 µg/L for ≥10 years)	420	289 (68.8)	221	146 (66.1)	199	143 (71.9)	0.2
Serum iron (µg/L)	420	894.9 (659.1–1110.1)	221	888.5 (672.7–1130.4)	199	910.1 (651.9–1100.8)	0.824
Serum iron <598 µg/L	420	81(19.3)	221	43 (19.5)	199	38 (19.1)	0.925
Serum ferritin, SF (ng/mL) ^$^	417	21.1 (11.3–33.7)	221	22.5 (12.2–35.6)	196	20.1 (11.2–33.5)	0.552
Storage iron deficiency (SF < 15 ng/mL)	417	147 (35.3)	221	68 (30.8)	196	79 (40.3)	0.042
Serum transferrin receptor, STFR (mg/L)	418	3.4 (2.9–4.0)	220	3.4 (2.8–4.0)	198	3.4 (3.0–4.1)	0.267
Functional iron deficiency (SFTR > 4.59 mg/L)	418	56 (13.4)	220	29 (13.2)	198	27 (13.6)	0.892
Iron deficiency (SF < 15 ng/mL or SFTR > 4.59 mg/L)	417	167 (40.0)	221	80 (36.2)	196	87 (44.4)	0.089
Haemoglobin, Hb (g/dL)	419	12.8 ± 1.2		12.9 ± 1.3	200	12.8 ± 1.1	0.715
Anaemia (Hb < 11.5 g/dL for < 12 years or < 12.0 g/dL for ≥12 years)	419	69.0 (16.5)	219	39 (17.8)	200	30 (15.0)	0.439
Iron Deficiency Anaemia (SF < 15 ng/mL or SFTR > 4.59 mg/L and Hb < 11.5 g/dL for <12 years or <12.0 g/dL for 12 years)	420	39 (9.3)	221	22 (10.0)	199	17 (8.5)	0.619
Anaemia Grade:	419		219		200		0.651
Mild (Hb = 11.0–11.4 g/dL for <12 years or 11.0–11.9 g/dL for ≥12 years)		43 (10.3)		23 (10.5)		20 (10.0)	
Moderate (Hb = 8.0–10.9 g/dL)		25 (6.0)		15 (6.8)		10 (5.0)	
Severe (Hb < 8.0 g/dL)		1 (0.2)		1(0.5)		0 (0.0)	
Non- anaemic (Hb ≥ 11.5 g/dL for <12 years or ≥12.0 g/dL for ≥12 years)		350 (83.5)		180 (82.2)		170 (85.0)	
Red blood cell count, RBC count (10^6^/µL)	420	4.6 ± 0.5	220	4.6 ± 0.5	200	4.6 ± 0.5	0.564
RBC count < 3.84 × 10^6^/µL	420	16 (3.8%)	220	7 (3.2%)	200	9 (4.5%)	0.481
Mean corpuscular volume, MCV (fL)	417	82.1 ± 7.0	217	82.0 ± 7.5	200	82.3 ± 6.4	0.714
MCV < 74.7 fL	417	41 (9.8%)	217	23 (10.6%)	200	18 (9.0%)	0.584
Haematocrit, HCT (%)	420	37.5 ± 3.0	220	37.5 ± 3.1	200	37.5 ± 2.8	0.857
HCT < 33.5%	420	33 (7.9%)	220	21 (9.5%)	200	12(6.0%)	0.177
Mean corpuscular haemoglobin concentration, MCHC (g/dL)	420	34.2 ± 1.3	220	34.2 ± 1.3)	200	34.1 ± 1.3	0.842
MCHC < 32.3 g/dL	420	34 (8.1)	220	18 (8.2)	200	16 (8.0)	0.946
C-reactive protein, CRP (mg/dL)	414	0.03 (0.02–0.07)	219	0.03 (0.02–0.06)	195	0.04 (0.02–0.08)	0.902
CRP >0.5 mg/dL	414	2 (0.5)	219	1 (0.5)	195	1(0.5)	0.934
Alpha 1-acid glycoprotein, AGP (mg/dL)	420	61.2 (45.5–74.5)	221	58.8 (43.3–71.6)	199	63.7 (48.2–79.6)	0.052
AGP > 100 mg/dL	420	14.0 (3.3)	221	4 (1.8)	199	10 (5.0)	0.067
Plasma selenium (µg/L)	418	102.8 ± 13.6	220	101.6 ± 12.4)	198	104.2 ± 14.7	0.020
Selenium deficiency (Plasma selenium < 41.8 μg/L)	418	0 (0.0)	220	0 (0.0)	198	0(0.0)	-
Plasma copper (µg/L)	420	930.4 ± 171.3	221	914.4 ± 183.0	199	948.1 ± 155.8	0.117
Copper deficiency (Plasma copper < 750 μg/L for < 10.3 years; <640 μg/L for 10.3–12.5 y; <570 μg/L for >12.5 y)	420	8 (1.9)	221	7 (3.2)	199	1 (0.5)	0.046
Copper:Zinc ratio	420	1.5 (1.3–1.7)	221	1.4 (1.2–1.7)	199	1.5 (1.3–1.7)	0.214

Data presented as Mean ± SD, Median (IQR), or *n* (%). All the haematological parameters were assessed at the beginning of the intervention phase (T3) of the study. * *p*-values obtained using linear mixed models adjusted for cluster effect to test differences between the groups for continuous variable and categorical variables by Pearson’s chi-squared test. Significance was set at *p* < 0.05. ^$^ Serum ferritin adjusted for inflammation [25].

**Table 3 nutrients-14-01657-t003:** Comparative summary of the mineral content for two varieties of wheat flour.

	Galaxy *n* = 75	Zincol-2016 *n* = 62	β (CI) *	t	*p*
Zinc (mg/kg)	17.0 ± 2.6	20.7 ± 5.6	3.696 (2.258, 5.134)	5.083	<0.001
	(16.4–17.6)	(19.2–22.1)			
Iron (mg/kg)	23.8 ± 4.7	26.8 ± 5.6	3.044 (1.293, 4.794)	3.439	0.001
	(22.6–24.9)	(25.4–28.2)			
Copper (mg/kg)	2.6 ± 0.9	2.9 ± 1.2	0.344 (−0.019, 0.707)	1.875	0.063
	(2.4–2.8)	(2.6–3.2)			
Selenium (µg/kg)	45.0 ± 13.2	51.3 ± 14.2	6.235 (1.610, 10.860)	2.666	0.009
	(42.0–48.1)	(47.7–54.9)			
Calcium (mg/kg)	326.4 ± 25.3	340.3 ± 22.9	13.924 (5.702, 22.146)	3.349	0.001
	(320.4–332.3)	(334.4–346.1)			
Phosphorus (g/kg)	2.2 ± 0.2	2.4 ± 0.2			
	(2.2–2.3)	(2.3–2.4)	0.169 (0.093, 0.245)	4.405	<0.001

Data outside of parenthesis indicates Mean ± SD. CI (95%) are presented within the parenthesis. * Values represent beta coefficient and 95% CI from linear regression models. *p* was set at < 0.05.

**Table 4 nutrients-14-01657-t004:** Consumption of wheat bread and corresponding zinc, iron and phytate intakes (*n* = 412).

	Wheat Bread Consumption (g/Day)	Zinc (mg/Day)			Iron (mg/Day)			Phytate (mg/Day)
		FCD *	Galaxy	Zincol-2016	FCD*	Galaxy	Zincol-2016	FCD *
Mean ± SD	541 ± 134	7.8 ± 2.0	6.9 ± 1.7	8.4 ± 2.1	12.3 ± 3.1	9.6 ± 2.4	10.9 ± 2.7	1584 ± 397.1
Median (Range)	525 (212–1114)	7.7 (3.0–16.2)	6.7 (2.7–14.2)	8.1 (3.3–17.3)	12.0 (4.6–25.4)	9.4 (3.8–19.9)	10.6 (4.3–22.4)	1548 (589–3270)

Analysis included only those participants for whom at least two 24 h dietary recalls were available. Galaxy was the standard variety used as control; Biofortified variety Zincol-2016 was the intervention. * FCD, Food Composition Database. Mineral and phytate intakes were calculated based on the Indian food composition database [32].

**Table 5 nutrients-14-01657-t005:** Zinc deficiency prevalence for all participants (*n* = 420) and plasma zinc concentration of subgroup with zinc deficiency at baseline (*n* = 146) over time by study arms.

	Time Points	*n*	Control	*n*	Intervention	X^2^	β (CI) *	t	*p*
**Zinc deficiency prevalence *, *n* (%)**									
	Baseline	221	146 (66.1)	199	143(71.9)	1.639			0.200
	Midpoint	213	159 (74.6)	192	141(73.4)	0.077			0.781
	Endline	214	129 (60.3)	188	121 (64.4)	0.709			0.400
**Plasma zinc concentration **, Mean ± SD**									
	Baseline	146	581.8 ± 54.5	143	581.9 ± 51.2				
	Midpoint	140	591.9 ± 69.1	136	584.2 ± 77.2		−2.563(−24.156, 19.030)	−0.242	0.810
	Endline	142	613.3 ± 79.8	134	619.3 ± 103.5		11.085 (−19.865, 42.035)	0.733	0.470

* All the participants included. ** Only those with zinc plasma levels below the age-specific IZiNCG cut-offs at baseline included [5]. * Values represent beta coefficient and 95% CI from linear regression models. *p*-values obtained using linear mixed models adjusted for cluster effect and baseline values to test differences between the groups for continuous variables. Categorical variables tested by Pearson’s chi-squared test. Significance was set at *p* < 0.05.

**Table 6 nutrients-14-01657-t006:** Biomarkers of Iron status by study arm at baseline, midpoint and endline.

Outcomes	Time Points	*n*	Control	*n*	Intervention	β (95%CI) *	X^2^	t	*p*
Serum iron (µg/L)	Baseline	221	888.5 (672.7–1130.4)	199	910.1 (651.9–1100.8)				
	Midpoint	213	877.2 (632.3–1104.3)	192	894.5 (676.0–1127.2)	33.539 (−43.606, 110.684)		0.879	0.385
	Endline	214	856.7 (610.9–1077.5)	188	828.1 (552.9–1060.6)	−28.302 (91.397, 34.793)		−0.921	0.365
Serum iron < 598 µg/L	Baseline	221	43 (19.5)	199	38 (19.1)		0.009		0.925
	Midpoint	213	45 (21.1)	192	35 (18.2)		0.535		0.465
	Endline	214	49 (22.9)	188	54 (28.7)		1.783		0.182
Serum Ferritin, SF (ng/mL) ^$^	Baseline	221	22.5 (12.2–35.6)	196	20.1 (11.2–33.5)				
	Midpoint	212	26.2 (15.2–40.5)	194	23.1 (14.2–33.8)	−3.525 (−8.254, 1.203)		−1.515	0.139
	Endline	213	17.4 (9.2–31.2)	188	17.8 (7.3–29.5)	−3.549 (−8.622, 1.523)		−1.418	0.165
Storage iron deficiency (SF <15 ng/mL)	Baseline	221	68 (30.8)	196	79 (40.3)		4.139		0.042
	Midpoint	212	49 (23.1)	194	50 (25.8)		0.389		0.533
	Endline	213	88 (41.3)	188	80 (42.6)		0.063		0.802
Serum Transferrin Receptor, STFR (mg/L)	Baseline	220	3.4 (2.8–4.0)	198	3.4 (3.0–4.1)				
	Midpoint	212	3.3 (2.9–3.9)	191	3.4 (3.0–4.0)	−0.005 (−0.241, 0.232)		−0.040	0.968
	Endline	210	3.2 (2.8–3.8)	187	3.3 (2.7–4.1)	0.112 (−0.178, 0.403)		0.779	0.440
Functional iron deficiency (SFTR > 4.59 mg/L)	Baseline	220	29 (13.2)	198	27 (13.6)		0.019		0.892
	Midpoint	212	26 (12.3)	191	19 (9.9)		0.544		0.461
	Endline	210	26 (12.4)	187	32 (17.1)		1.775		0.183
Iron deficiency (SF < 15 ng/mL or SFTR > 4.59 mg/L)	Baseline	221	80 (36.2)	196	87 (44.4)		2.901		0.089
	Midpoint	209	60 (28.7)	192	57 (29.7)		0.046		0.829
	Endline	213	90 (42.3)	188	88 (46.8)		0.839		0.360
Iron Deficiency anaemia (SF <15 ng/mL or SFTR > 4.59 mg/L and Hb < 11.5 g/dL for <12 years or <12.0 g/dL for *≥*12 years)	Baseline	221	22 (10.0)	199	17 (8.5)		0.248		0.619
	Midpoint	213	16 (7.5)	195	13 (13.9)		0.110		0.740
	Endline	213	23 (10.8)	188	25 (13.3)		0.592		0.442

Data presented as Mean ± SD, Median (IQR), or *n* (%). ^$^ Ferritin adjusted for inflammation [25]. * Values represent beta coefficient and 95% CI from linear regression models. *p*-values obtained using linear mixed models adjusted for cluster effect and baseline values to test differences between the groups for continuous variables. Categorical variables tested by Pearson’s chi-squared test. Significance was set at *p* < 0.05.

**Table 7 nutrients-14-01657-t007:** Plasma copper and selenium concentration, and inflammation markers by study arm at baseline, midpoint and endline.

Outcomes	Time Points	*n*	Control	*n*	Intervention	β (95%CI) *	X^2^	t	*p* *
Plasma copper (µg/L)	Baseline	221	914.4 ± 183.0	199	948.1 ± 155.8				
	Midpoint	213	926.3 ± 139.3	191	961.3 ± 154.5	16.882 (−7.130, 40.894)		1.447	0.160
	Endline	214	871.0 ± 147.3	187	907.9 ± 155.9	18.562 (−10.056, 47.180)		1.326	0.195
Copper deficiency (plasma copper < 750 μg/L for <10.3 years; <640 μg/L for 10.3–12.5 years; <570 μg/L for >12.5 years)	Baseline	221	7 (3.2)	199	1 (0.5)		3.980		0.046
	Midpoint	213	2 (0.9)	191	1 (0.5)		0.236		0.627
	Endline	214	10 (4.7)	187	2 (1.1)		4.464		0.035
Plasma selenium (µg/L)	Baseline	220	101.6 ± 12.4	198	104.2 ± 14.7				
	Midpoint	213	98.5 ± 13.1	192	101.4 ± 12.7	0.960 (−2.637, 4.556)		0.536	0.595
	Endline	213	95.7 ± 12.1	188	96.0 ± 13.4	−0.270 (−3.483, 2.943)		−0.170	0.866
Copper:zinc ratio	Baseline	221	1.4 (1.2–1.7)	199	1.5 (1.3–1.7)				
	Midpoint	213	1.5 (1.3–1.7)	191	1.6 (1.4–1.8)	0.032 (−0.026, 0.090)		1.116	0.273
	Endline	214	1.4 (1.2–1.6)	187	1.4 (1.3–1.6)	0.024 (−0.032, 0.081)		0.876	0.388
C-reactive protein, CRP (mg/dL)	Baseline	219	0.03 (0.02–0.06)	195	0.04 (0.02–0.08)				
	Midpoint	213	0.030 (0.02–0.05)	194	0.03 (0.02–0.06)	0.005 (−0.014, 0.023)		0.535	0.597
	Endline	212	0.03 (0.02–0.05)	186	0.03 (0.02–0.07)	0.005 (−0.011, 0.021)		0.603	0.552
CRP > 0.5 mg/dL	Baseline	219	1 (0.5)	195	1 (0.5)		0.007		0.934
	Midpoint	213	0 (0.0)	194	2 (1.0)		2.207		0.137
	Endline	212	1 (0.5)	186	1 (0.5)		0.009		0.926
Alpha 1-acid glycoprotein, AGP (mg/dL)	Baseline	221	58.8 (43.3–71.6)	199	63.7 (48.2–79.6)				
	Midpoint	214	66.5 (55.6–80.5)	194	66.4 (56.3–79.7)	0.552 (−4.001, 5.106)		0.246	0.807
	Endline	214	69.5 (57.5–83.0)	188	63.7 (52.8–76.8)	−4.548 (−8.921, −0.175)		−2.148	0.042
AGP > 100 mg/dL	Baseline	221	4 (1.8)	199	10 (5.0)		3.359		0.067
	Midpoint	214	12 (5.6)	194	17 (8.8)		1.534		0.215
	Endline	214	14 (6.5)	188	5 (2.7)		3.350		0.067

No deficiency of selenium observed (Plasma selenium < 41.8 μg/L). Data presented as Mean ± SD, Median (IQR), or *n* (%). * Values represent beta coefficient and 95% CI from linear regression models. *p*-values obtained using linear mixed models adjusted for cluster effect and baseline values to test differences between the groups for continuous variables. Categorical variables tested by Pearson’s chi-squared test. Significance was set at *p* < 0.05.

**Table 8 nutrients-14-01657-t008:** Spearman correlation coefficient between plasma copper and selenium concentrations and inflammatory markers.

		Copper	Selenium	C-Reactive Protein	Alpha-1-Acid Glycoprotein	Copper:Zinc Ratio
Copper	r_s_	1.000	0.166 **	0.216 **	0.252 **	0.744 **
	*n*	2169	2160	2123	2163	2168
Selenium	r_s_	0.166 **	1.000	0.062 **	−0.001	−0.005
	*n*	2160	2165	2119	2158	2159
C-Reactive Protein	r_s_	0.216 **	0.062 **	1.000	0.456 **	0.189 **
	*n*	2123	2119	2154	2150	2122
Alpha-1-Acid Glycoprotein	r_s_	0.252 **	−0.001	0.456 **	1.000	0.222 **
	*n*	2163	2158	2150	2193	2162
Copper:Zinc ratio	r_s_	−0.573 **	−0.086 **	0.222 **	0.090 **	1.000
	*n*	2168	2153	2162	2152	2168

** *p*-value < 0.01. r_s_, Spearman correlation coefficient.

## Data Availability

The dataset of this study can be made available on reasonable request to Nicola M. Lowe.

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
