# Peer review of "The Impact of Consuming Zinc-Biofortified Wheat Flour on Haematological Indices of Zinc and Iron Status in Adolescent Girls in Rural Pakistan: A Cluster-Randomised, Double-Blind, Controlled Effectiveness Trial"

_nutrients, 2022, doi:10.3390/nu14081657_

Round 1

Reviewer 1 Report

Why phytic acid was not measured in this study? it is know that phytic acid affects zinc absorption.

Considering that the differential of zinc content between biofortified and control flours was only 3.3 mg/kg, it will be relevant to add discussion about other common extraction rates in wheat flours in Pakistan (besides the 80% used), regardless of the sieving procedures at home. 

Lines 248-249 indicate that phytate content of the wheat flours was measured. However, it appears that such results were not presented. Table 4 presented phytate content but it indicated that the values are from a Food Composition Database. It is relevant to discuss the difference in phytate content of the flours used.

Lines 881-885. Add reference.

Lines 889-902. Add reference.

Reviewer 2 Report

This well designed study analyzes the effectiveness of Zn-biofortified wheat flour consumption on zinc and iron status in adolescent girls in Pakistan. In this population group, the prevalence of zinc deficiency based on plasma zinc concentration is a serious public health problem. It is well known that plasma zinc concentration is only reduced in moderate or severe zinc deficiency. This indicator does not detect mild zinc deficiencies.

The fact that the consumption of Zn-biofortified wheat flour for 5.5 months did not significantly reduce the prevalence of zinc deficiency could be due to several factors: A daily amount of absorbed zinc less than that required to cover the needs of the studied group and/or a shorter duration of the intervention than is necessary to observe an effect. Another possibility is a compliance failure to the consumption of zinc-biofortified wheat flour. Although the compliance was evaluated, no information is given on the results obtained. This information is important to rule out a failure to comply with the intervention. All These facts should be included or deepened in the manuscript.

Table 2. Was the normality of the distribution of the laboratory parameters assessed? Several of the laboratory parameters have a standard deviation that suggests a skewed distribution (serum iron, serum ferritin, serum transferrin receptor, C-reactive protein, alpha 1-acid glycoprotein, copper:zinc ratio). If any of them has a skewed distribution, it should be expressed as a median with another measure of dispersion (i.e. interquartile range).

Table 8. Did any of the correlations have a p-value <0.001? (see footnote of the table)
